# Salinity Effects on Morpho-Physiological and Yield Traits of Soybean (*Glycine max* L.) as Mediated by Foliar Spray with Brassinolide

**DOI:** 10.3390/plants10030541

**Published:** 2021-03-13

**Authors:** Victoria Otie, Idorenyin Udo, Yang Shao, Michael O. Itam, Hideki Okamoto, Ping An, Egrinya A. Eneji

**Affiliations:** 1Department of Soil Science, Faculty of Agriculture, Forestry and Wildlife Resources Management, University of Calabar, Calabar P.M.B. 1115, Nigeria or victoriaotie@unical.edu.ng (V.O.); or anthonyeneji@unical.edu.ng (E.A.E.); 2Arid Land Research Centre, Tottori University, Hamasaka, Tottori 680-0001, Japan; D16A4001Z@edu.Tottori-u.ac.jp; 3Department of Crop Science, Faculty of Agriculture, Forestry and Wildlife Resources Management, University of Calabar, Calabar P.M.B. 1115, Nigeria; idyudo@gmail.com or; 4United Graduate School of Agricultural Sciences, Tottori University, Tottori 680-8553, Japan; itammichaelo@gmail.com; 5Agriculture Research Department, Hokkaido Research Organization, Hokkaido 060-0817, Japan; okamoto-hideki@hro.or.jp

**Keywords:** *Glycine max* (L.), saline stress, brassinolide, antioxidants, arid region, dryland

## Abstract

Salinity episodes that are common in arid regions, characterized by dryland, are adversely affecting crop production worldwide. This study evaluated the effectiveness of brassinolide (BL) in ameliorating salinity stress imposed on soybean at four levels (control (1.10), 32.40, 60.60 and 86.30 mM/L NaCl) in factorial combination with six BL application frequency (control (BL_0_), application at seedling (BL_1_), flowering (BL_2_), podding (BL_3_), seedling + flowering (BL_4_) and seedling + flowering + podding (BL_5_)) stages. Plant growth attributes, seed yield, and N, P, K, Ca and Mg partitioning to leaves, stems and roots, as well as protein and seed-N concentrations, were significantly (*p* ≤ 0.05) reduced by salinity stress. These trends were ascribed to considerable impairments in the photosynthetic pigments, photosynthetically active radiation, leaf stomatal conductance and relative water content in the leaves of seedlings under stress. The activity of peroxidase and superoxidase significantly (*p* ≤ 0.05) increased with salinity. Foliar spray with BL significantly (*p* ≤ 0.05) improved the photosynthetic attributes, as well as nutrient partitioning, under stress, and alleviated ion toxicity by maintaining a favourable K^+^/Na^+^ ratio and decreasing oxidative damage. Foliar spray with brassinolide could sustain soybean growth and seed yield at salt concentrations up to 60.60 mM/L NaCl.

## 1. Introduction

Abiotic stresses such as salinity, heat, drought, acidity, and waterlogging cause severe damages to crop production. Soil salinity (a major threat to crop production across the globe) affects various plant physiological activities through increased oxidative damage, decreased turgor, as well as changes in leaf gas exchange [1], ultimately leading to reduced plant growth, development and yield [2,3]. In arid and semi-arid regions of the world, crop productivity has been impaired by salt accumulations [4]. According to the Food and Agricultural Organization (FAO) [5], saline soils affect 6% of the world’s land mass, equivalent to 831 million hectares. Salinity decreases the external osmotic potential in soil solution, causing ion accumulations and reactive oxygen species (ROS) production [6]. Salt (NaCl) damage can cause the necrosis of older leaves, enhancing Na^+^ influx and K^+^ leakage that leads to an elevated Na^+^/K^+^ ratio in plant cells [7,8]. Increased salinity alters the plants’ morphology, physiology and metabolism through increased ion toxicity that leads to decreases in the availability of essential elements such as phosphorus and calcium, as well as protein synthesis and metabolism of lipids [9]. These result in premature senescence and the loss of photosynthetic efficiency that lead to reductions in carbon assimilation and yield. Although plants have the capacity to generate various forms of antioxidants that can alleviate oxidative damages, there is a need to further investigate other exogenous applications, including the use of phytoprotectants such as brassinolide (BL), to boost plants’ oxidative defence against salinity stress.

Brassinolide is a plant growth regulator that plays diverse roles in cell division and expansion, stem elongation, xylem differentiation, and root growth [10]. It plays mitigating roles in plants under drought stress [11,12], soil acidity [13], waterlogging [14,15], heat [16] and heavy metals [17]. Brassinolide was also reported to ameliorate the adverse effects of salt on stomatal conductance, membrane permeability and leaf water content, as well as ionic composition in salt-stressed sheep grass and strawberries [18,19]. Fariduddin et al. [17] found that BL can influence protein synthesis, improve the plasma membrane, increase the uptake and assimilation of nutrients, and further facilitate the efficient translocation of photosynthetic products from source to sink under salt stress conditions.

Soybean (*Glycine max* L.) is one of the most important leguminous crops in the world, grown for its edible oil and forage. It has been considered to have a moderate tolerant threshold of about 5.0 dS/m of saturated soil extract [20]. According to An et al. [21], the tolerant species of soybean have better growth tendencies than the susceptible ones due to their root differences. Roots of the tolerant species absorb less salt than the sensitive ones [22]. Ghassemi-Golezani et al. [23] also reported that soil moisture stress inhibits faba bean production. Similarly, a high salinity reduces soybean height, leaf size, biomass, number of branches, internodes, pods and seed yield significantly [24]. Many studies have shown how BL could induce salt tolerance in rice [25], strawberries [18] and *Arabidopsis thaliana* [26], but little is known of its role in ameliorating salinity stress on soybean at various growth stages. In this study, we explored the effectiveness of BL on ameliorating the salinity effect on soybean at morphological and physiology levels, as well as yield responses.

## 2. Materials and Methods

### 2.1. Site Trial Management

The experiment was conducted in the Glass House of the subdivision of Plant Eco-Physiology, Arid Land Research Center (ALRC), Tottori University, Tottori-Japan (35°31′7″ N, 134°12′54.5″ E). Tottori soils are mainly sand-dune soils. The soil used for this study was collected at three different points from the dune site and bulked. A composite sample (0–15 cm depth) from the site was analysed for routine physical and chemical properties using standard laboratory procedures [27,28].

The treatments studied were four levels of salinity, control (tap water (1.10 mM/L)), 32.40, 60.60 and 86.30 mM/L NaCl in combination with six application frequencies of BL (control (no application), application at seedling (when the trifoliolate leaves unfold), flowering (when one flower opens at any node on the main stem), podding (when pods begin to emerge on one of the four uppermost nodes on the main stem), seedling + flowering and seedling + flowering + podding growth stages). The salinity levels were achieved by dissolving 1.75, 3.51 and 5.26 g salt (NaCl), respectively, in 1 litre (L) of distilled water. The corresponding concentrations read from an electrical conductivity meter were 3.24 (32.40), 6.06 (60.60) and 8.63 dS/m (86.30 mM/L). The treatments were arranged into a 4 × 6 factorial in a completely randomized design (CRD) with five replications, giving a total of 24 treatment combinations and 120 experimental units. Liquid-undiluted nutrient solution (NPK HYPONEX (6:10:5)) produced by Kabushiki Corporation, Tokyo-Japan was used across the pots as a base nutrient source. The BL (24-epibrassinolide) was obtained from XenanXinyu Chemical Technology Company Limited (Ltd), Henan Province, China. Sodium chloride (NaCl) was procured from Gourmet Meat World Company Ltd., Tottori, Japan, while the soybean seeds (*Tachiyutaka*8428h) were obtained from Denki nojo, Yamagata-Japan. Seeds of soybean were surface-sterilized in 5% hypochlorite solution for 10 mins and thoroughly rinsed thrice with distilled water before sowing. Plastic pots (height, 30 cm; diameter, 25 cm), perforated at the base for drainage, were filled with 20 kg of the sandy soil. Three seeds were sown per pot and later thinned to one vigorous plant at 10 days after sowing (DAS). The liquid NPK fertilizer was diluted at the rate of 5–1000 mL distilled water and applied (200 mL per pot) at 2 and 5 weeks after sowing (WAS). Tap water was used to irrigate plants daily during the first two weeks of sowing. The saline treatment commenced at 3 WAS by irrigating plants with the saline water (500 mL) according to the treatment combinations each time until the end of the experiment. The BL was dissolved following the manufacturer’s recommended rate of 1 g to 20 L distilled water and foliar sprayed (200 mL) per plant at 20 days after sowing (DAS) (seedling stage), 30 DAS (flowering stage) and 40 DAS (podding stage), using a calibrated hand sprayer.

The electrical conductivity of soil extract (EC_se_) was determined on a weekly basis, using an electrical conductivity meter (CM-20E, TOA, Tokyo TOA Electronics Ltd., Japan) to monitor salt concentrations in the rhizosphere. Cultural practices like hand weeding and insecticide spray-Sumition (Sumitomo Chemical Garden Product Incorporated, Tokyo-Japan) were done at 3 and 6 WAS.

### 2.2. Observations

Plant data collection was carried out bi-weekly. The data collected were:

#### 2.2.1. Growth Variables

Plant height, number of leaves and leaf area index (LAI) were measured bi-weekly. The specific leaf area (SLA) was determined as the ratio of leaf area of soybean to dry weight of leaves per plants at 6 WAS.

#### 2.2.2. Nutrient Analysis

The shoot biomass (leaves, stems and seeds) and roots were sampled at harvest and oven-dried at 80 °C to a constant weight for 72 h. Dry samples of leaves, stems and roots were ground using a shaker device (DWB5K-D1807-5, Tottori, Japan). The ground samples (40 mg) were soaked in 10 mL of 1% HCl at room temperature for 30 min. The extract was centrifuged at 15,000 rpm for 15 min at 20 °C. The supernatant was collected, filtered and analysed for potassium (K), magnesium (Mg), calcium (Ca) and sodium (Na), using the atomic absorption spectrophotometer (AA 6800, ASC-61, Shimadzu Corporation, Japan). The K^+^/Na^+^ ratio was also determined. Dry soybean seeds were milled with a cell homogenizer shake master device (BMS-12, Japan) at 1100 rpm for 10 min to analyse the N concentration, using the micro-Kjeldahl method [29]. The protein content in soybean seeds was also determined by multiplying the total nitrogen (TN) concentration with a soybean factor of 5.71 [30].

#### 2.2.3. Plant Physiological Measurements

The photosynthetically active radiation (PAR) was determined as a spot measurement on three locations of the youngest mature leaves, using a Quantum Integral Sensor (MQ-100: Apogee Instrument INC, Logan, Utah-USA) and averaged at 20, 30 and 40 days after sowing (DAS). To determine the effect of salinity on plant photosynthesis, a handheld Minolta chlorophyll meter (SPAD 502 plus, KONICA MINOLTA- Tottori, Japan) was used. The SPAD values were also obtained from three locations of the youngest mature leaves and averaged at 20, 30 and 40 DAS. Stomatal conductance was similarly estimated, using the leaf porometer (AP4 No.1, DELTA-T DEVICES-Cambridge, United Kingdom). The relative water content in leaves (RWCL) was calculated from the fresh, turgid and dry weight of leaves according to Smart and Bingham [31]:RWCL (%) = (Fresh weight-Dry weight)/(Turgid weight-Dry weight) × 100(1)

#### 2.2.4. Biological Yield and Yield Components

The harvest index, leaf and stem dry weights, shoot-to-root ratio and seed yield (g) per plant were recorded at harvest. The number of branches, pods and seeds per plant were also determined at harvest. The number of days to 50% flowering and podding was also recorded.

#### 2.2.5. Enzyme Extractions and Assays

Fully expanded young leaves (0.5 g) of soybean plants were sampled at 25 DAS, at four BL application levels (control (no application), seedling, flowering, and seedling + flowering growth stages of soybean) and two salinity levels (1.10 and 60.60 mM/L). Samples were frozen in liquid nitrogen and ground in 5 mL Tris buffer solution containing 0.25 M sucrose, 10 mM Tris, and 1 mM EDTA at pH 7.4. The homogenate was centrifuged at 4800 rpm for 15 min at 4 °C. The supernatant was collected for enzyme assays.

The activity of superoxide dismutase (SOD) was assayed using the SOD Assay Kit-WST (Dojindo Molecular Technologies Incorporated, Tottori, Japan) following the method described in [32]. The reaction plate was incubated in a micro-plate reader (SH-9000, Corona Electronics, Ibaraki, USA) at 37 °C for 20 min. The absorbance of each reaction mixture was read at 450 nm.

Peroxidase (POD) activity was assayed using the POD Assay Kit-WST (Elabscience-E-BC-K227-S, Houston, Texas-USA). One gram (1 g) of leaf tissue was homogenized with PBS (0.01 M, pH 7.4). Homogenized samples were centrifuged at 10,000× *g* for 10 min at 4 °C. The reaction samples were incubated at 37 °C for 30 min. The reaction solution was mixed thoroughly and centrifuged at 2300× *g* for 10 min. The optical density value of each sample was measured spectrophotometrically (ASCO-V-560, Tottori, Japan) at 420 nm with a 1 cm optical path cuvette.

The ascorbate peroxide activity (APX) was assayed using the APX Colorimetric Assay Kit-WST (Elabscience-E-BC-K353-S, Houston, TX, USA). Leaf tissues (0.5 g) ground in liquid nitrogen were homogenized in 5 mL Tris extraction buffer. The homogenate was centrifuged at 10,000× *g* for 10 min at 4 °C. The APX was measured at 290 nm for 15 s (A1), and the reaction solution was incubated at 37 °C and measured for 135 s (A2), using a spectrophotometer (ASCO-V-560, Japan).

### 2.3. Statistics

The data were subjected to analysis of variance using GenStat software 15.1 Edition to partition the effects of the two factors and their interactions. Treatment means were compared using Duncan’s new multiple-range test at a 5% level of probability.

## 3. Results

### 3.1. Soil Analysis

The analysis of the experimental soil showed it to be dominated by sand with 961 g/kg, followed by clay (350 g/kg) and silt (40 g/kg). It was classified as coarse-textured Entisols, characterized by weak surface aggregation. The soil had a pH (H_2_O) of 5.98, and contained (g/kg) 0.1 total nitrogen, 0.3 organic carbon, and 0.6 available P; (cmol/kg) 0.06 exchangeable K, 0.35 exchangeable Ca, and 0.68 exchangeable Mg. The soil was also characterized by low (2.0 meq/100 g) cation exchange capacity (CEC), indicating its inherently poor fertility status.

### 3.2. Plant Growth

The interaction between salinity and BL application was significant (*p* ≤ 0.05) for all the growth variables evaluated at the different sampling periods (Table 1). Soybean plants under the highest salt concentration (S_3_ = 86.30 mM/L), with no application of BL, were significantly (*p* ≤ 0.05) shortest, while those grown without salt stress (S_0_ = 1.10 mM/L) and with BL applied at BL_5_ had significantly (*p* ≤ 0.05) the tallest plants across sampling periods. At 6 WAS, BL application at flowering (BL_2_) to plants under the highest salt concentration (S_3_) significantly (*p* ≤ 0.05) produced the tallest plants, while at 8 WAS, at the same salt concentration, BL_1_ (seedling stage) and BL_4_ produced the tallest plants.

The trend of interaction effects for the number of leaves and leaf area index (LAI) at all sampling periods was similar to that of plant height. At the highest salinity level, BL_4_ significantly (*p* ≤ 0.05) produced the highest number of leaves at 6 WAS, while at 8 WAS, BL_1_ and BL_4_ produced the highest number of leaves per plant (Table 1). For LAI, there were no significant changes with BL application at the highest salt concentration whether at 6 or 8 WAS (Table 1). Plants under BL_5_ had the largest (*p* ≤ 0.05) SLA across salinity levels. At any stage of BL application, successive increases in salinity level led to a significant reduction in SLA. Plants under the highest level of salinity without BL had the smallest SLA, while those without salt stress (S_0_) and sprayed at BL_5_ had the largest SLA

### 3.3. Photosynthetically Active Radiation

The interaction effects of salinity and BL (S × BL) on photosynthetically active radiation (PAR) at 20, 30 and 40 days after sowing (DAS) were significant (Figure 1a–c). At 20 DAS, with no salt stress (1.10 mM/L), BL application (except at BL_3_) significantly enhanced PAR relative to no application. The same trend was observed at the 32.40 mM/L salt level. However, at 60.60 mM/L salt concentration, BL had little effect on PAR, except when applied at BL_4._ At the highest salt level, the BL significantly enhanced the PAR, especially at BL_4_ and BL_5_ (Figure 1a); a similar trend was noted at 30 DAS (Figure 1b). At 40 DAS, increases in salinity reduced the PAR significantly, but BL significantly enhanced it across salinity levels. However, without salt (S_0_), and at the highest salt level (S_3_), the application at BL_5_ best (*p* ≤ 0.05) improved the PAR (Figure 1c).

### 3.4. Relative Water Content in Leaves

Figure 2a,b show the interaction effect of salinity and BL on the relative water content in leaves (RWCL) at 20 and 30 DAS. At 20 DAS, BL application at BL_3_ had a negligible effect on the RWCL at S_0_ and S_2_ salt levels. However, it significantly enhanced the RWCL at different salinity levels with the highest under BL_5_ (Figure 2a). At 30 DAS, increases in salinity significantly reduced the RWCL, but BL, especially BL_4_ and BL_5_, significantly enhanced it across salinity levels (Figure 2b). The RWCL of stressed plants at 40 DAS increased significantly following BL application (Figure 3a), especially at BL_5_. Increases in salinity levels significantly reduced the RWCL of the plants (Figure 3b).

### 3.5. Chlorophyll Concentration in Leaves

At 20 DAS, with or without salt stress, BL application significantly increased the chlorophyll concentration in leaves. Generally, there were little changes in chlorophyll concentration among plants sprayed with BL at different growth stages (Figure 4a). At 30 and 40 DAS, increases in salinity levels led to a significant fall in chlorophyll concentration. Across salt levels, BL significantly enhanced the chlorophyll concentration, and for control plants without stress, BL_5_ best (*p* ≤ 0.05) improved the concentration (Figure 4b,c).

### 3.6. Leaf Stomatal Conductance

The leaf stomatal conductance was progressively (*p* ≤ 0.05) reduced with increases in salinity level at the different sampling periods (20, 30 and 40 DAS) (Figure 5a–c). At 20 DAS, BL increased (*p* ≤ 0.05) the stomatal conductance, especially at BL_5_ (Figure 5a). The trend of response was similar at 30 and 40 DAS. Brassinolide application at BL_4_ and BL_5_ best enhanced stomatal conductance across salinity levels (Figure 5b,c).

### 3.7. Superoxide Dismutase (SOD), Peroxidase (POD) and Ascorbate Peroxidase (APX)

The interaction effects of salinity and BL on antioxidant enzyme production at the 50% flowering stage (25 DAS) are shown in Figure 6. There was a considerable (*p* ≤ 0.05) increase in SOD and POD activity with the exposure of plants to salinity at all application stages of BL (Figure 6a,b). Brassinolide increased the SOD activity, with or without salinity. Plants at BL_3_ (seedling + flowering stages) had the highest (*p* ≤ 0.05) SOD activity (Figure 6a). Plants not exposed to salinity had a lower POD activity. In addition, salt-stressed plants sprayed with BL had similar POD activity, except at BL_1_ (seedling stage), which showed a much higher (*p* ≤ 0.05) activity than the control plants (BL_0_) (Figure 6b). The interaction effect of salinity and BL application was not significant (*p* > 0.05) on the APX activity (Figure 6c) of soybean plants.

### 3.8. Calcium [Ca^2+^], Potassium [K^+^], Magnesium [Mg^2+^], Sodium [Na^+^] and K^+^/Na^+^ Ratio

The concentration of Ca in leaves and stems decreased significantly with successive increases in the concentration of salt (Table 2), and the root concentration under salinity was much (p ≤ 0.05) lower than in control plants. Similarly, higher levels of salinity (60.60 to 86.30 mM/L) were averse to K and Mg concentrations of the below and above-ground biomass. However, it significantly increased the concentration of Na in the above and below-ground parts (Table 2). The K^+^/Na^+^ ratio was much (p ≤ 0.05) higher in both the above- and below-ground partitions of control plants than those of salt-stressed plants. Generally, BL significantly increased the concentrations of Ca, K and Mg in the above- and below-ground plant partitions. However, there was little difference in the root concentration of Ca between the treated (BL_1_ to BL_5_) and the untreated plants (BL_0_). The BL_1_, BL_4_ and BL_5_ treatments significantly reduced the leaf concentration of Na, but for the stem, only BL_5_ caused a significant reduction. Root concentrations of Na were substantially reduced following BL treatment. The K^+^/Na^+^ ratio of the leaf and stem were the same with or without BL application; however, BL_5_ significantly increased the K^+^/Na^+^ ratio in the roots compared with BL_0_ and other stages of BL application (Table 2).

### 3.9. Leaf Dry Weight (LDW), Stem Dry Weight (SDW), Harvest Index (HI), Shoot-to-Root Ratio (S:R) and Protein Concentration

Successive increases in salt concentration led to significant (*p* ≤ 0.05) decreases in LDW, SDW, S:R and seed protein (Table 3). The HI was substantially reduced at the higher salt levels (S_2_ and S_3_) relative to the control, but BL significantly increased it with or without salt stress. The LDW and SDW were best improved at BL_3_, while BL_5_ produced seeds with the highest protein concentration and S:R (Table 3).

There was a significant interaction effect of salinity and BL on LDW and SDW. For both variables, BL application under S_0_ (1.10 mM/L) and S_1_ (32.40 mM/L) significantly enhanced their yield but not under higher salt levels (S_2_ = 60.60 mM/L and S_3_ = 86.30 mM/L) (Table 3). Under control conditions (S_0_ = 1.10 mM/L), BL_4_ significantly increased the HI and BL_5_ best increased (*p* ≤ 0.05) the S:R. At the highest salinity level (S_3_), plants that received no BL had the lowest (*p* ≤ 0.05) S:R; however, the S:R decreased progressively with salt concentration, with or without BL (Table 3).

### 3.10. Number of Branches/Plant, Days to 50% Flowering and Podding, Number of Pods and Seeds/Plant

Successive increases in salt concentration led to a corresponding significant decrease in the number of branches per plant (Table 4). However, BL application increased it significantly at BL_5_ compared with no application. Increases in salinity levels significantly delayed flowering, while higher salinity (60.60 and 86.30 mM/L) significantly delayed podding. Brassinolide application irrespective of the growth stage significantly reduced the number of days to 50% flowering and podding. Irrespective of BL application, the highest salt concentrations significantly increased the number of days to 50% flowering and podding (Table 4). Plants under no or low salt stress (32.40 mM/L) and treated with BL showed significant earliness in flower and pod production. The number of pods and seeds per plant were adversely affected by salinity, but the BL significantly increased these yield attributes, especially at BL_5_ under stress (Table 4).

### 3.11. Nitrogen (N) Concentration and Seed Yield

The nitrogen concentration in soybean seeds varied inversely with salt concentration (Figure 7a). Except at the highest concentration (86.30 mM/L), BL application significantly increased seed-N concentration. Without salt stress, BL_5_ significantly produced seeds with the highest N concentration. Conversely, a negligible seed-*N* concentration was observed at the highest level of salinity. At S_1_ and S_2_ salt concentrations, all stages of BL application had a similar effect on seed-N concentration. The seed yield was considerably reduced at higher salt concentrations (S_2_ and S_3_) and BL had no significant (p > 0.05) effect (Figure 7b). However, control plants that received BL_3_, BL_4_ and BL_5_ and those exposed to the lowest salinity (S_1_ = 32.40 mM/L) showed much (*p* ≤ 0.05) higher seed yield. Plants that received no BL at the highest salt concentration (S3 = 86.30 mM/L) had zero yield (Figure 7b).

## 4. Discussion

Salinity stress is a significant constraint to the growth and development of soybean plants in arid regions. With the adoption of plant-salt resistance at various growth stages of soybean, through the use of BL, a plant growth regulator may promote yield under this stress. It has been suggested previously [33] that a crop’s growth stage is an important aspect to consider in BL application under stressed soil conditions.

### 4.1. Plant Growth, Harvest Index and Biomass

Growth attributes (plant height, number of leaves, leaf area index (LAI) and specific leaf area (SLA)) were, as expected, adversely affected by salinity stress, especially at the highest salinity concentration (S_3_ = 86.30 mM/L). This is possibly due to the restriction of cell elongation and division that may lead to the inhibition of physiological and biochemical processes in the plants [34]. The reduction in growth attributes such as shoot length, number of leaves, LAI and SLA due to salinity has been reported in [35,36]. The exogenous spraying of BL improved soybean growth under salinity stress. This is ascribed to its ability to limit salt toxicity, by modulating cell division and elongation, as well as the differentiation of stem cells in root meristem, thereby promoting root growth [37]. However, the best growth attributes were recorded when BL was applied to control plants at the seedling + flowering + podding growth stages (BL_5_). This positive effect of BL is also linked to its ability to regulate a wide range of processes, including source–sink relationships [38].

There were significant reductions in the HI and S:R ratio of plants as salinity level increased. The decrease in economic yield is attributed to the decline in yield components (number of pods and seeds/plant) and seed yield due to salt stress and their associated adverse effects on physiological processes [39]. Salinity-induced reductions in the S:R ratio of leguminous plants have been previously documented [40]. The application of BL enhanced the HI and S:R ratio across soybean growth stages, especially at the BL_4_ and BL_5_ growth stages. Brassinolide could facilitate biological yield and yield-related traits because of its ability to detoxify salinity stress induced by NaCl, through the modification of various gas exchange parameters [41,42].

A salt-induced decrease in biomass production was identified with a reduction in leaf and stem dry weight of soybean plants. According to [36], salt stress inhibits cell division and elongation, thus reducing plant growth and biomass potentials. In addition, an impairment of essential nutrient uptake could have accounted for a decrease in plant biomass of the salt-stressed plants [43]. Exogenously applied BL produced the highest leaf and stem dry weight at the BL_3_ (podding stage) stage of plants under the control (S_0_ = 1.10 mM/L) or mild salinity (S_1_ = 32.40 mM/L) stress. Yang et al. [44] reported that BL could improve photosynthetic activities by enhancing pigmentation and osmotically mediated adjustments in plants for increased biomass.

The agronomic traits of soybean in this study were severely affected by salinity stress. This includes the number of branches, number of pods and seeds/plant, which may be attributed to the inability of the plants to form new branches and the early senescence of older branches due to salt stress [45]. Taffouo et al. [46] attributed the decrease in the number of pods and seeds/plant of cowpea to reductions in leaf chlorophyll concentrations. The application of BL significantly enhanced these yield components, especially at the BL_5_ growth stage, possibly because of its ability to degrade older cell walls and promote new cell wall biosynthesis to boost growth and development [19]. The number of days to 50% flowering and podding was similarly prolonged by salinity stress at the higher salinity levels (S_2_ = 60.60 mM/L and S_3_ = 86.30 mM/L). Reduced crop production at increasing salinity levels is a product of ionic imbalance that causes toxicity in plants [47]. The number of days to 50% flowering and pod development was significantly shortened by BL application, as reported previously [48,49]. Brassinolide spraying enhanced early flowering and podding mainly due to its ability to activate the plants’ defence system (antioxidants) against stress effects for increased yield and yield components [50].

### 4.2. Stomatal Conductance, Chlorophyll Concentration and Photosynthetically Active Radiation

A reduction in stomatal conductance in soybean leaves was observed at various levels of salinity. This consistent salt-induced decrease was a result of stomatal closure due to the direct effect of the stress on photosynthetic capacity, such as diffusion of carbon dioxide (CO_2_) into the leaves [51]. Stomatal conductance was greatly improved when BL was applied at the BL_4_ and BL_5_ growth stages, suggesting that the BL enabled CO_2_ diffusion to improve stomatal variables and leaf anatomy [52,53], thereby minimizing the negative effects of salinity.

Leaf chlorophyll concentration is an indicator of chloroplast formation, photosynthetic efficiency and the general well-being of plants. In this study, salinity had adverse effects on photosynthetic pigments, thereby causing a severe reduction in leaf chlorophyll across sampling periods. Hayat et al. [54] reported that salt stress reduced the concentration of leaf chlorophyll, and affected gas exchange tendencies of *Vigna radiata.* Similarly, a recent study [35] showed that salinity caused a degradation of photosynthetic pigment synthesis in tomatoes, characterized by lower chlorophyll and carotenoid contents. However, in this study, salt-stressed soybean was able to improve chlorophyll formation with exogenous supplementation with BL. Brassinolide spraying at 30 and 40 DAS enhanced the leaf chlorophyll content across salinity levels, especially in control plants. This may be ascribed to the improved production of photosynthetic pigments and gas exchange potentials, as well as regulations of chlorophyllase activities in the plants [55].

The photosynthetically active radiation (PAR) was severely disrupted by salinity stress, possibly through the alterations of stomatal activities or decline in photosynthetic efficiency [56]. In this study, increases in salinity levels aggravated the stress severity across the sampling periods through reductions in LAI, chlorophyll content and stomatal conductance. Rahman et al. [57] showed that salinity stress elevated the chlorophyllase activity, causing a reduction in chlorophyll formation and leading to leaf senescence and chlorosis. The application of BL increased the PAR, with or without salt stress, possibly by regulating the combination of chlorophyll molecules with membrane protein, and stabilizing the thylakoid membranes [55], thus reducing their concentrations under salt stress. Xuan and Khang [58] also reported how chlorophyll and carotenoid were improved in tomatoes treated with exogenous vanillic acid under salinity stress. At 20 and 30 DAS, BL enhanced PAR across salinity levels, especially at the BL_3_ (podding), BL_4_ (seedling + flowering) and BL_5_ (seedling + flowering + podding) stages. This is an indication of the potential protective role of BL against stress imposed by salinity [19].

### 4.3. Relative Water Content in Leaves

Under saline conditions, plants suffer from osmotic stress due to an imbalance in osmotic pressure in plant cells. More so, salt accumulation in root zones reduces the root water potential and restricts water-uptake, which may cause osmotic imbalances and decrease the biomass. Successive increases in salinity concentrations decreased the relative water content in leaves (RWCL) significantly. It was previously reported [36] that salt-induced osmotic stress lowered the RWCL and elevated proline contents in tomato. However, BL spraying boosted the RWCL across sampling periods, especially at the BL_4_ and BL_5_ application stages. This could be attributed to its unique ability to antagonize any deficit in water-uptake by promoting and improving membrane stability and plants’ physiological mechanisms against stress imposed by salinity [59,60].

### 4.4. Superoxide Dismutase (SOD) Activity

In stressed plants, the antioxidant defence system is one of the most important options for reactive oxygen species (ROS) regulation. According to Hasanuzzaman et al. [61], some enzymatic antioxidants like the SOD, ascorbate peroxidase (APX), catalase (CAT), glutathione peroxidase (GPX) and glutathione *S*-transferase (GST) are important enzymatic components for controlling ROS in plants under stress conditions. In this study, salinity increased SOD activities. This is consistent with recent reports [43] where salt toxicity augmented SOD activity and induced an extreme generation of ROS in plants. Conversely, BL application boosted SOD activities, with or without salt stress, especially when applied at the seedling + flowering stages. This could be ascribed to the reduction in O^-^_2_ and H_2_O_2_ induced by BL, thus elevating the antioxidant enzyme activities in the plant system for better salt tolerance [62]. Similarly, Xuan and Khang [58] and Singh et al. [35] reported that exogenous supplementation with vanillic acid increased the activities of SOD and CAT, by suppressing H_2_O_2_ accumulation in salt-treated seedlings, and also subdued ROS in rice under submergence conditions.

### 4.5. Peroxidase (POD) Activity

The activity of POD in leaves increased significantly under salinity stress as previously reported [63]. Brassinolide application best increased POD of soybean leaves at the BL_1_ (seedling) growth stage, relative to other stages. The enhanced activity following BL spray could be due to its ability to counter the harmful reactive oxygen species (ROS) formed under stressful conditions. A recent report by Yadava et al. [64] showed that BL enhanced the antioxidant system in maize plants (*Zea mays* L.) under heat-stress.

### 4.6. Ascorbate Peroxidase (APX) Activity

There were no significant effects of the treatments, whether singly or in combination, on the ascorbate peroxidase (APX) activity of soybean plants. Interestingly, increasing the levels of salinity and the use of BL did not alter APX activity. This contrasts a report by Hasanuzzaman et al. [65] that BL spray could improve APX through the reduction in H_2_O_2_ under oxidative stress. Kumutha et al. [61] found that remarkable enzymatic components, APX-inclusive, could control ROS under salinity stress.

### 4.7. Mineral [Calcium (Ca^2+^), Potassium (K^+^), Magnesium (Mg^2+^), Sodium (Na^+^)] Contents and K^+^/Na^+^ Ratio

Salt-induced toxicity (Na^+^) causes a reduction in mineral elements, including K^+^, Ca^2+^ and Mg^2+^. The report of Rahman et al. [57] confirmed that this reduction was caused by a hindrance in homeostasis that leads to ionic stress. For all the soybean plant partitions (leaves, stems and roots), K^+^, Ca^2+^ and Mg^2+^ uptake was adversely affected by salt stress as observed in previous studies [8,66]. However, Na^+^ increased significantly across plant partitions due to salt stress. This excessive Na^+^ uptake may have led to an imbalance in cellular Na^+^ and K^+^ contents. The metabolic activities of the plants’ cytoplasm and organelles were significantly disturbed by the accumulations of Na^+^ in shoot and root tissues [67]. A lower K^+^/Na^+^ ratio was noted across plant partitions under salinity than in control plants. As salt stress results in ion imbalance, the need for exclusion of extra Na^+^ and Cl^-^ is necessary for plant survival. The relative decrease in K^+^/Na^+^ due to salinity stress has been previously reported [68,69].

The exogenous spraying of BL improved the uptake of K^+^, Ca^2+^ and Mg^2+^ across the plant partitions. Under salinity stress, BL has the ability to modify the plant plasma membrane and increase nutrient uptake and assimilation, as well as translocation of photosynthates to the sink [70]. The concentration of Na^+^ in soybean leaves was significantly reduced at the BL_1_, BL_4_ and BL_5_ application stages of BL. In the stem, a decrease in Na^+^ was observed at the BL_5_ stage, whereas all BL application stages reduced [Na^+^] significantly in soybean roots. The ability of BL to maintain the structure of the plasma membrane may have contributed to the significant decrease in Na^+^ and Cl^−^ ions, and the enhancement of K^+^, Ca^2+^ and Mg^2+^ ions [71]. The K^+^/Na^+^ ratio was highest in roots at the BL_5_ growth stage. Recent reports have revealed that BL prevents NaCl-induced K^+^ leakage from shoots and roots [72] and boosts the K^+^/Na^+^ ratio in roots and leaves, thus alleviating Na^+^ toxicity [73].

### 4.8. Protein and Nitrogen Concentrations

Adequate N uptake improves the mobilization and growth of plants; protein is an important constituent of soybean seeds. Successive increases in the levels of salinity decreased the contents of protein and N in soybean seeds, and at the highest salt level studied (S_3_ = 86.30 mM/L), the seed-N was zero. The consistent decrease in seed-N concentration due to salinity may be associated with a reduction in nitrate absorption [74]. The inadequate N may also be a result of an alteration in intracellular ion homeostasis that impaired the plant’s ability to take up N ions for translocation to the leaves, as well as carbon metabolisms due to protein degradation [75]. Brassinolide spray at the BL_5_ growth stage boosted protein and N concentrations in soybean seeds. This is consistent with earlier reports [16,25,76].

### 4.9. Seed Yield

Seed yield is an important agricultural goal that enhances food security, and as expected, the soybean yield was significantly reduced by salinity stress, with no yield at all at the highest salinity stress (S_3_ = 86.30 mM/L). This reduction was due to the high level of toxic Na^+^ in soybean leaves that led to premature leaf senescence and defoliation, thereby inhibiting the supply of photosynthates, as well as flower and pod abortion. A reduction in soybean yield under salinity stress has been reported previously by Mannan et al. [48]. The application of BL improved seed yield under stress, but the best yield was obtained in control plants sprayed at BL_3_ and BL_5_ without salt and those under mild salt stress (S_1_ = 32.40 mM/L). The remarkable improvement in seed yield following BL application may be due to its ameliorative role in stress tolerance and the reinforcement of the plant’s immunity for reproductive development [77,78]. From the seed yield result, it could be inferred that soybean (or at least, the cultivar-*Tachiyutaka*8428h studied) cannot tolerate high salinity stress up to 86.30 mM/L, even with BL application, as no seed was obtained. The poor seed yield was directly attributed to reduced growth and impairment of physiological and reproductive traits. However, exogenous spray with BL alleviated the deleterious effects of salinity to boost soybean production when applied at the BL_5_ (seedling + flowering + podding) growth stage. This validates the claim that multiple applications of BL at different growth stages could confer some tolerance on soybean plants under salinity stress.

In conclusion, salinity adversely affected the growth, dry matter accumulation, photosynthetic activities, nutrient uptake and partitioning, other physiological attributes and seed yield of soybean cv. *Tachiyutaka8428h.* Exogenous foliar spray of 24-epibrassinolide at various growth stages ameliorated the above-stated adverse effects by maintaining a favourable ionic balance (k^+^/Na^+^), enhanced antioxidative enzyme activities and increased seed yield relative to no BL application. Thus, the application of BL at seedling + flowering + podding growth stages could be used to mitigate the damaging effects of salt on soybean production, but further study is advocated to validate the efficiency of the optimal BL application stage (BL_5_) under natural field conditions.

## Figures and Tables

**Figure 1 plants-10-00541-f001:**
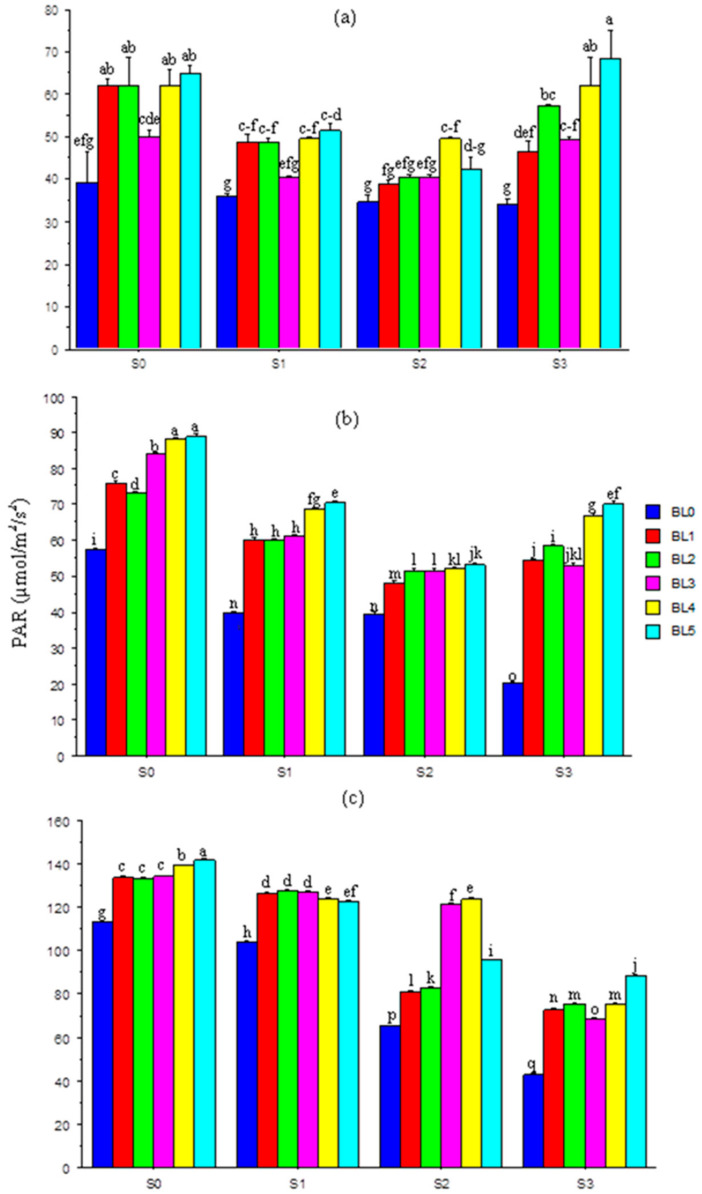
Interaction effects of salinity × brassinolide on PAR of soybean at 20 (**a**), 30 (**b**) and 40 (**c**) DAS. PAR—photosynthetically active radiation; BL—brassinolide; S—salinity; DAS—days after sowing; BL0—no BL application (control); BL1—BL application at seedling; BL2—BL application at flowering; BL3—BL application at podding; BL4—BL application at seedling + flowering; BL5—BL application at seedling + flowering + podding; S0—1.10 mM/L (control); S1—32.40 mM/L; S2—60.60 mM/L; S3—86.30 mM/L. Mean pairs with different letters are significantly different at the 5% probability level according to Duncan’s new multiple-range test.

**Figure 2 plants-10-00541-f002:**
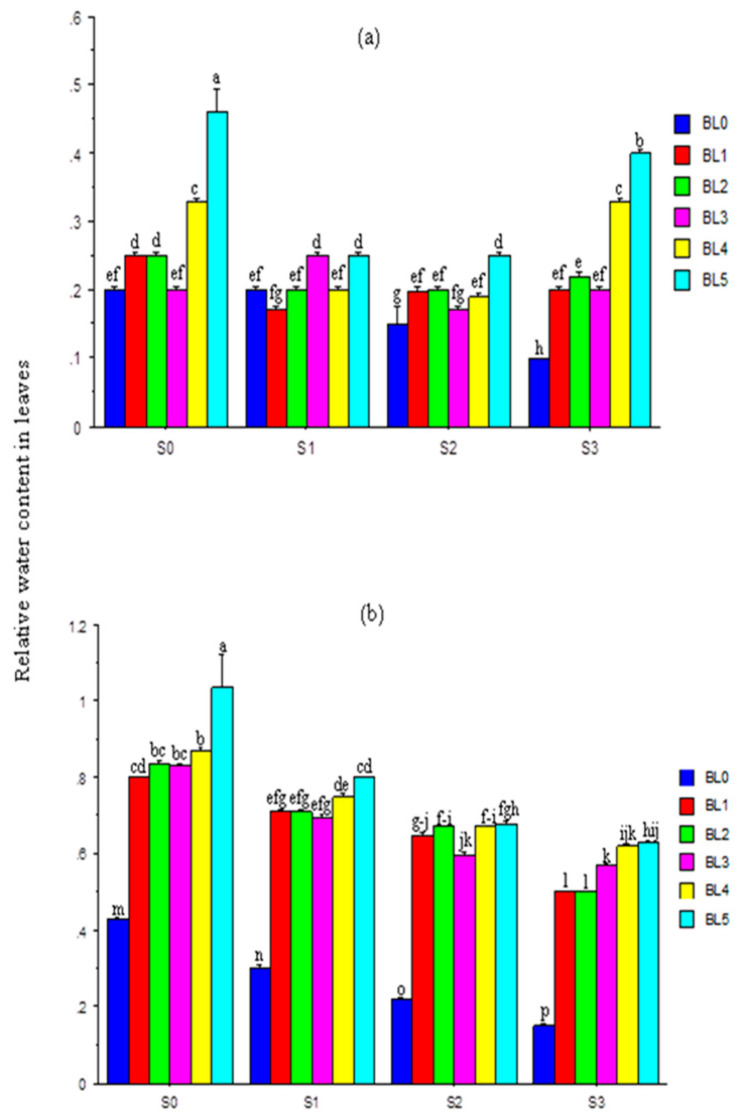
Interaction effects of salinity × brassinolide on relative water content in leaves of soybean at 20 (**a**) and 30 (**b**) DAS. BL—brassinolide; S—salinity; DAS—days after sowing; BL0—no BL application (control); BL1—BL application at seedling; BL2—BL application at flowering; BL3—BL application at podding; BL4—BL application at seedling + flowering; BL5—BL application at seedling + flowering + podding; S0—1.10 mM/L (control); S1—32.40 mM/L; S2—60.60 mM/L; S3—86.30 mM/L. Mean pairs with different letters are significantly different at the 5% probability level according to Duncan’s new multiple-range test.

**Figure 3 plants-10-00541-f003:**
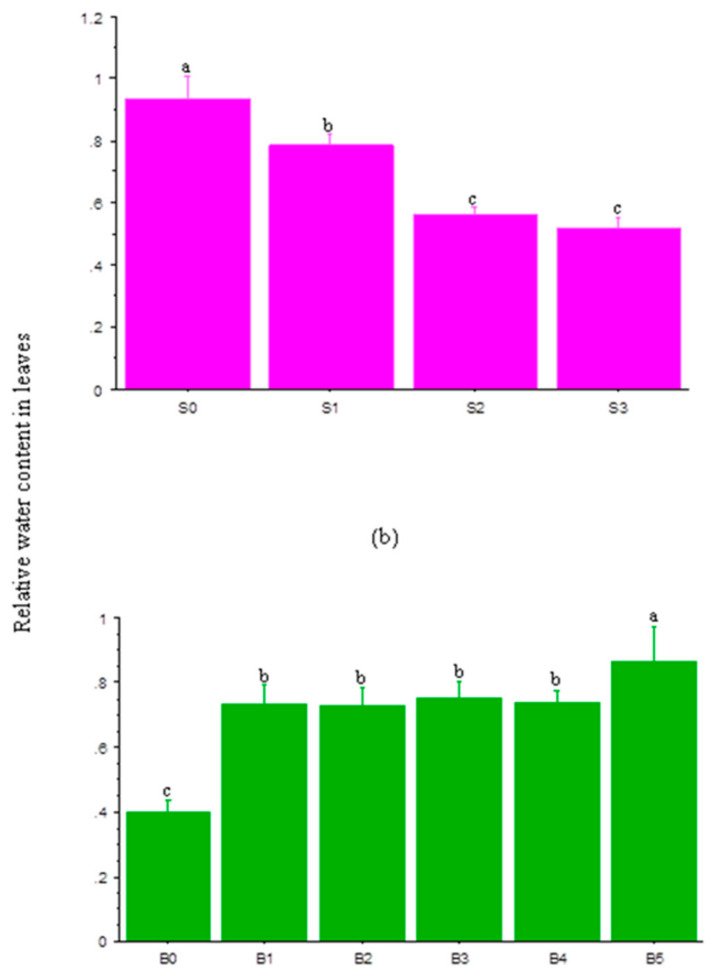
Single effects of salinity (**a**) and brassinolide (**b**) on relative water content in leaves at 40 DAS. BL—brassinolide; S—salinity; DAS—days after sowing; B0—no BL application (control); BL1—BL application at seedling; BL2—BL application at flowering; BL3—BL application at podding; BL4—BL application at seedling + flowering; B5—BL application at seedling + flowering + podding; S0—1.10 mM/L (control); S1—32.40 mM/L; S2—60.60 mM/L; S3—86.30 mM/L. Mean pairs with different letters are significantly different at the 5% probability level according to Duncan’s new multiple-range test.

**Figure 4 plants-10-00541-f004:**
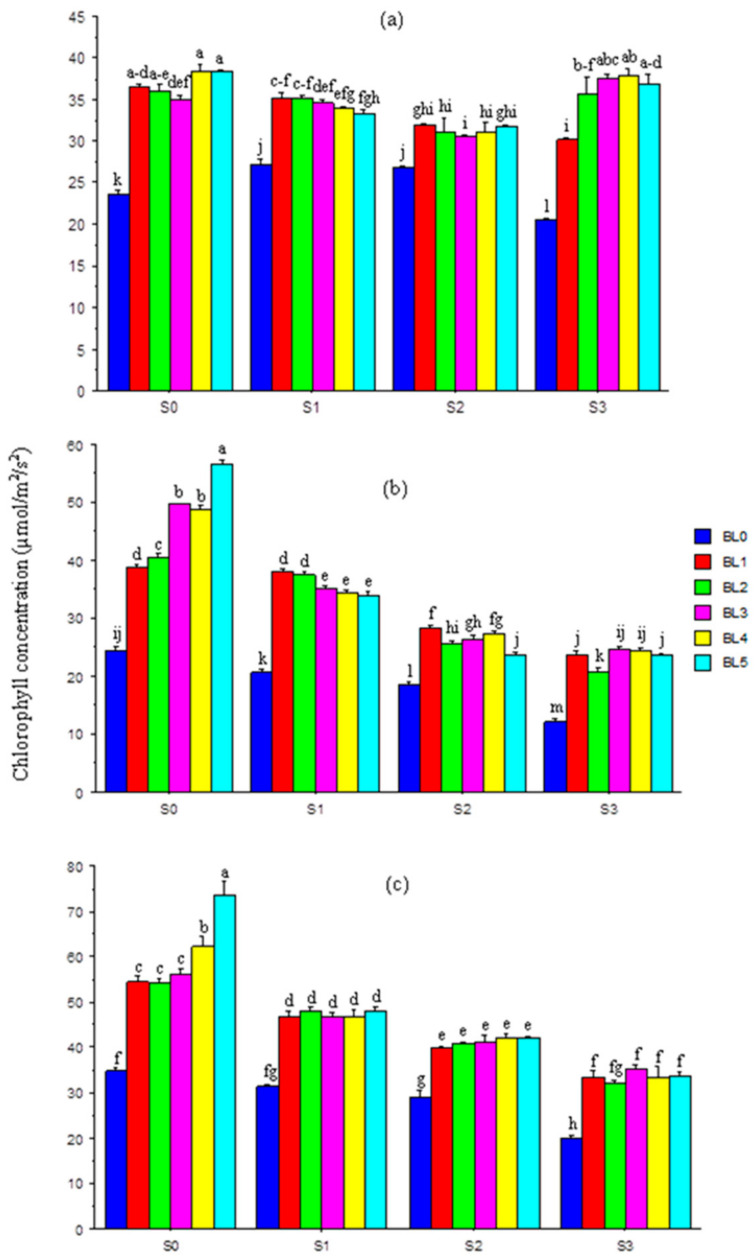
Interaction effects of salinity × brassinolide on chlorophyll content of soybean at 20 (**a**), 30 (**b**) and 40 (**c**) DAS. BL—brassinolide; S—salinity; DAS—days after sowing; BL0—no BL application (control); BL1—BL application at seedling; BL2—BL application at flowering; BL3—BL application at podding; BL4—BL application at seedling + flowering; BL5—BL application at seedling + flowering + podding; S0—1.10 mM/L (control); S1—32.40 mM/L; S2—60.60 mM/L; Scheme 3. 30 mM/L. Mean pairs with different letters are significantly different at the 5% probability level according to Duncan’s new multiple-range test.

**Figure 5 plants-10-00541-f005:**
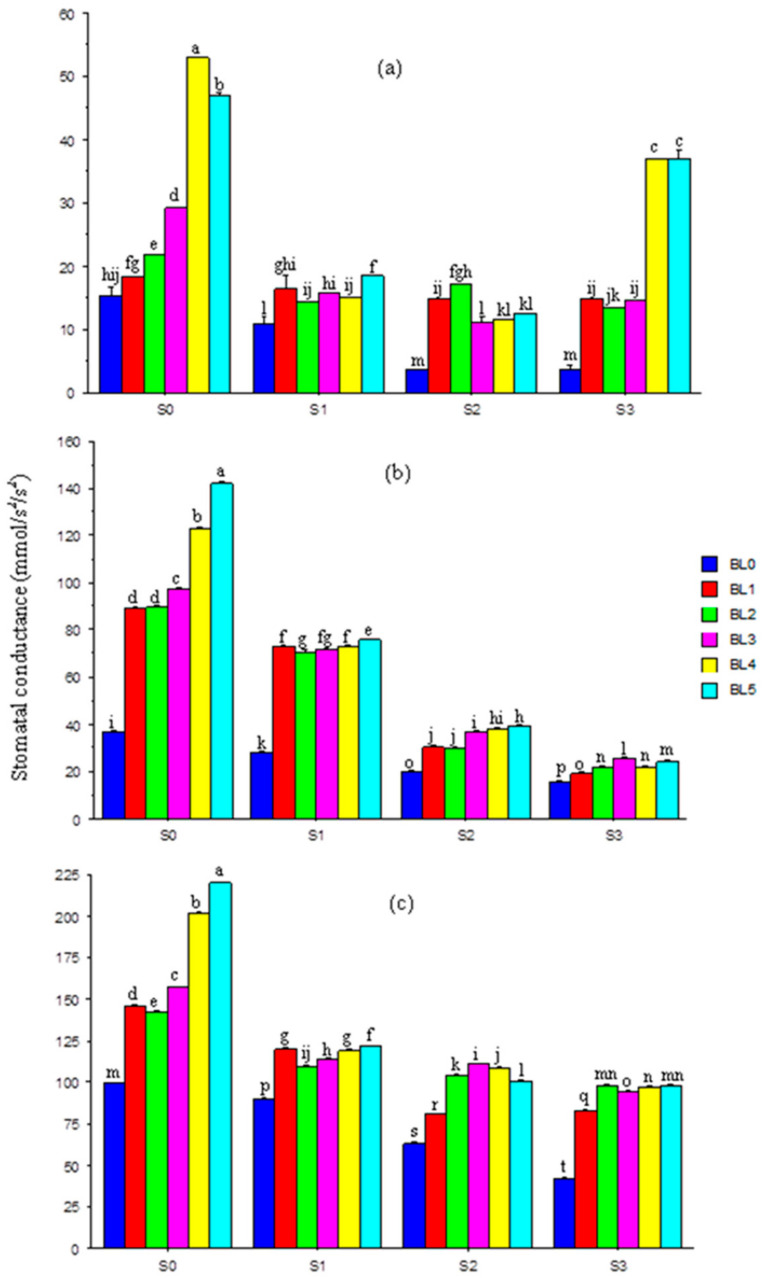
Interaction effects of salinity × brassinolide on stomatal conductance of soybean at 20 (**a**), 30 (**b**) and 40 (**c**) DAS. BL—brassinolide; S—salinity; DAS—days after sowing; BL0—no BL application (control); BL1—BL application at seedling; BL2—BL application at flowering; BL3—BL application at podding; BL4—BL application at seedling + flowering; BL5—BL application at seedling + flowering + podding; S0—1.10 mM/L (control); S1—32.40 mM/L; S2—60.60 mM/L; S3—86.30 mM/L. Mean pairs with different letters are significantly different at the 5% probability level according to Duncan’s new multiple-range test.

**Figure 6 plants-10-00541-f006:**
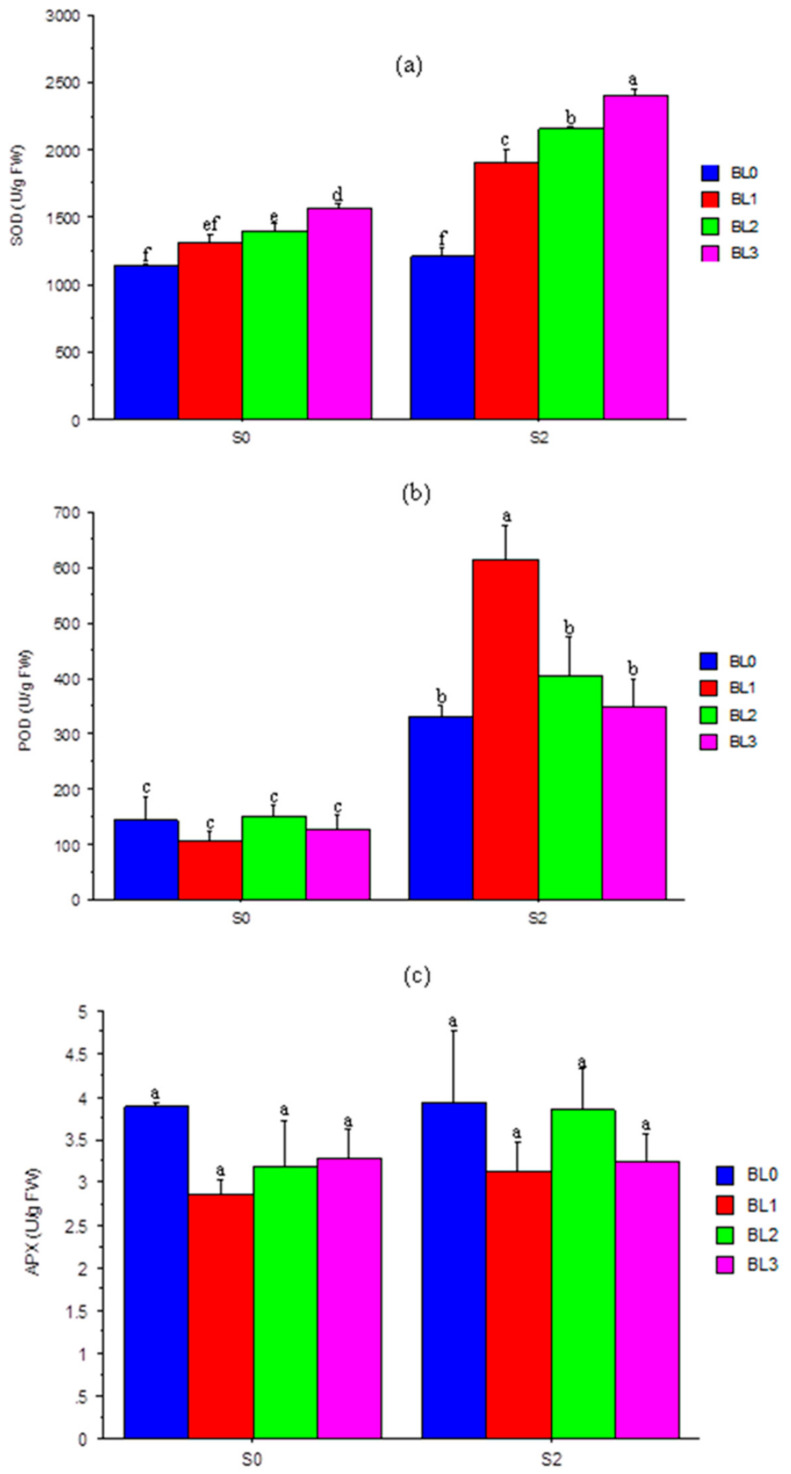
Interaction effects of salinity × brassinolide on the activity of SOD (**a**), POD (**b**) and APX (**c**) of soybean leaves at flowering growth stage. SOD—superoxide dismutase; POD—peroxidase; APX—ascorbate peroxide; BL—brassinolide; Scheme 0. BL application at seedling; BL2—BL application at flowering; BL3—BL application at seedling + flowering; S0—1.10 mM/L (control); S2—60.60 mM/L. Mean pairs with different letters are significantly different at the 5% probability level according to Duncan’s new multiple-range test.

**Figure 7 plants-10-00541-f007:**
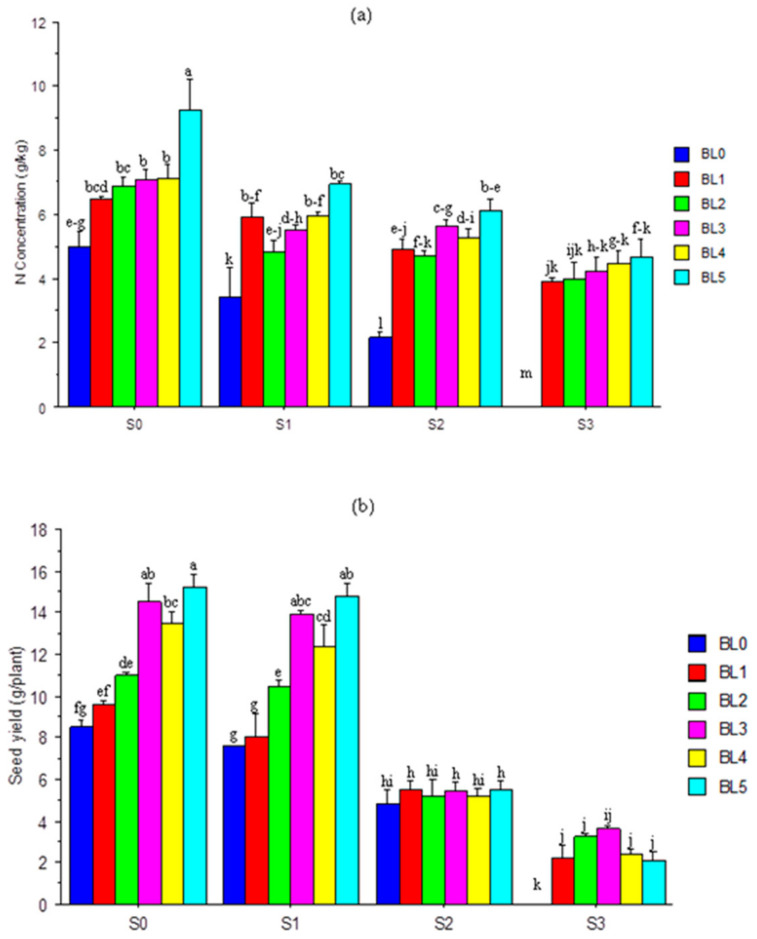
Interaction effects of salinity × brassinolide on seed-*N* concentration (**a**) and yield of soybean (**b**) at harvest. BL—brassinolide; S—salinity; BL0—no BL application (control); BL1—BL application at seedling; BL2—BL application at flowering; BL3—BL application at podding; BL4—BL application at seedling + flowering; BL5—BL application at seedling + flowering + podding; S0—1.10 mM/L (control); S1—32.40 mM/L; S2—60.60 mM/L; S3—86.30 mM/L. Mean pairs with different letters are significantly different at the 5% probability level according to Duncan’s new multiple-range test.

**Table 1 plants-10-00541-t001:** Interaction effects of salinity (S) and brassinolide (BL) on plant height, number of leaves, leaf area index and specific leaf area of soybean at different sampling periods.

Sources of Variance	Plant Height (cm)				Number of Leaves				Leaf Area Index (LAI)			Specific Leaf Area (cm^2^/g)
**S × BL**	2 WAS	4 WAS	6 WAS	8 WAS	2 WAS	4 WAS	6 WAS	8 WAS	4WAS	6WAS	8WAS	6WAS
**S_0_BL_0_**	14.33 ^efg^	21.00 ^j^	40.15 ^ef^	48.90 ^bcd^	7.00 ^de^	17.33 ^j–m^	30.33 ^l^	38.67 ^gh^	2.91 ^gh^	4.64 ^fgh^	5.67 ^def^	152.40 ^l^
**S_1_ BL_0_**	13.67 ^efg^	17.99 ^k^	41.10 ^ef^	42.73 ^e–i^	10.00 ^b^	13.67 ^n^	30.00 ^l^	36.67 ^h–k^	2.40 ^hi^	3.81 ^ghi^	4.45 ^ghi^	142.40 ^mn^
**S_2_ BL_0_**	14.00 ^efg^	18.25 ^k^	30.82 ^m^	37.87 ^i^	9.00 ^b–e^	10.46 ^o^	21.00 ^l^	31.67 ^klm^	2.13 ^i^	2.78 ^ij^	3.12 ^j^	133.50 ^p^
**S_3_ BL_0_**	11.67 ^g^	13.34 ^l^	20.10 ^n^	24.57 ^j^	8.00 ^cde^	6.00 ^p^	12.00 ^n^	20.00 ^p^	1.07 ^j^	2.38 ^j^	3.13 ^j^	119.50 ^q^
**S_0_ BL_1_**	13.00 ^fg^	34.70 ^c^	43.37 ^d^	66.53 ^a^	8.00 ^cde^	23.33 ^cd^	58.00 ^e^	69.33 ^c^	4.36 ^dc^	6.38 ^bcd^	7.70 ^c^	208.10 ^de^
**S_1_ BL_1_**	13.00 ^fg^	34.94 ^c^	41.50 ^e^	44.33 ^d–h^	14.00 ^a^	19.33 ^f–l^	39.00 ^gh^	41.67 ^fg^	3.95 ^ef^	5.59 ^def^	5.91 ^def^	196.80 ^g^
**S_2_ BL_1_**	16.00 ^cde^	31.52 ^de^	38.07 ^gh^	40.57 ^f–i^	8.00 ^cde^	17.00 ^j–n^	26.00 ^j^	34.33 ^h–l^	2.75 ^ghi^	4.73 ^efg^	5.31 ^efg^	166.80 ^k^
**S_3_ BL_1_**	17.00 ^bcd^	23.62 ^i^	32.53 ^l^	44.87 ^d–g^	8.00 ^cde^	19.67 ^e–k^	20.00 ^l^	33.00 ^i–m^	2.67 ^ghi^	3.59 ^g–j^	4.52 ^ghi^	137.30 ^o^
**S_0_ BL_2_**	12.00 ^g^	40.13 ^b^	46.70 ^c^	66.73 ^a^	5.00 ^f^	24.00 ^c^	64.00 ^d^	71.33 ^c^	3.70 ^ef^	6.40 ^bcd^	8.64 ^b^	209.30 ^d^
**S_1_ BL_2_**	14.00 ^efg^	31.26 ^de^	41.77 ^e^	45.57 ^d–g^	8.00 ^cde^	20.33 ^d–j^	37.00 ^h^	46.37 ^e^	3.86 ^ef^	5.29 ^def^	5.78 ^def^	195.90 ^g^
**S_2_ BL_2_**	14.00 ^efg^	28.98 ^fg^	34.63 ^k^	38.70 ^hi^	9.00 ^b–e^	17.00 ^i–n^	27.00 ^j^	33.00 ^i–m^	2.90 ^gh^	5.47 ^def^	5.42 ^ef^	151.70 ^l^
**S_3_ BL_2_**	12.00 ^g^	25.14 ^hi^	39.47 ^fg^	42.87 ^e–i^	9.00 ^b–e^	21.00 ^c–f^	18.00 ^m^	24.67 ^no^	2.61 ^ghi^	4.55 ^fgh^	5.01 ^f–i^	135.50 ^op^
**S_0_ BL_3_**	16.00 ^cde^	38.77 ^b^	46.80 ^c^	69.40 ^a^	10.00 ^b^	28.00 ^b^	69.00 ^c^	72.33 ^c^	4.71 ^cd^	6.88 ^bc^	9.07 ^b^	215.50 ^c^
**S_1_ BL_3_**	8.00 ^h^	32.04 ^de^	43.63 ^d^	46.77 ^cde^	8.00 ^cde^	21.67 ^c–f^	39.00 ^gh^	46.00 ^ef^	3.30 ^fg^	5.94 ^cde^	6.10 ^de^	196.40 ^g^
**S_2_ BL_3_**	15.00 ^def^	28.34 ^g^	37.33 ^hi^	39.87 ^ghi^	7.00 ^de^	17.00 ^h–n^	26.00 ^j^	31.33 ^lm^	2.93 ^gh^	4.75 ^efg^	5.13 ^fgh^	174.70 ^i^
**S_3_ BL_3_**	13.00 ^fg^	28.89 ^fg^	36.30 ^ij^	39.67 ^ghi^	8.00 ^cde^	15.67 ^mn^	18.00 ^m^	22.67 ^op^	2.73 ^ghi^	3.78 ^g–j^	4.40 ^hi^	140.40 ^n^
**S_0_ BL_4_**	17.00 ^bcd^	40.14 ^b^	53.57 ^b^	67.43 ^a^	8.00 ^cde^	28.33 ^b^	72.00 ^b^	81.67 ^b^	5.53 ^b^	7.51 ^b^	9.33 ^b^	227.30 ^b^
**S_1_ BL_4_**	15.00^def^	30.42 ^ef^	43.51 ^d^	51.90 ^bc^	8.00 ^cde^	22.33 ^c–f^	38.00 ^gh^	44.00 ^ef^	3.94 ^ef^	5.54 ^def^	6.20 ^de^	201.40 ^f^
**S_2_ BL_4_**	12.00 ^g^	26.18 ^h^	40.77 ^ef^	46.10 ^def^	8.00 ^cde^	20.33 ^d–j^	24.00 ^k^	34.00 ^h–l^	2.58 ^hi^	3.35 ^hij^	4.25 ^hi^	170.70 ^j^
**S_3_ BL_4_**	19.00^ab^	24.78 ^hi^	32.28 ^lm^	44.17 ^d–h^	7.00 ^de^	16.33 ^k–m^	24.00 ^k^	31.67 ^j–m^	2.77 ^ghi^	3.72 ^ghi^	4.12 ^i^	141.90 ^mn^
**S_0_ BL_5_**	20.33 ^a^	42.44 ^a^	56.67 ^a^	68.93 ^a^	8.00 ^cde^	32.00 ^a^	78.33 ^a^	88.00 ^a^	6.42 ^a^	9.25 ^a^	11.80 ^a^	233.40 ^a^
**S_1_ BL_5_**	15.00 ^def^	33.10 ^cd^	43.46 ^d^	53.07 ^b^	8.00 ^cde^	23.00 ^cde^	47.00 ^f^	52.67 ^d^	5.03 ^bc^	5.54 ^def^	6.34 ^d^	206.20 ^e^
**S_2_ BL_5_**	18.00 ^abc^	26.44 ^h^	35.63 ^jk^	39.93 ^ghi^	8.00 ^cde^	22.67 ^c–f^	23.00 ^k^	36.67 ^h–k^	2.94 ^gh^	3.43 ^hij^	4.29 ^hi^	188.60 ^h^
**S_3_ BL_5_**	17.67 ^bc^	24.82 ^hi^	34.53 ^k^	40.83 ^f–i^	8.00 ^cde^	16.00 ^lmn^	20.00 ^l^	29.00 ^mn^	5.57 ^hi^	3.77 ^ghi^	4.25 ^hi^	143.40 ^m^

Mean pairs within a column with different letters are significantly different at the 5% probability level according to Duncan’s new multiple-range test. BL_0_—no brassinolide application (control); BL_1_—brassinolide application at seedling growth stage of soybean; BL_2_—brassinolide application at flowering growth stage of soybean; BL_3_—brassinolide application at podding growth stage of soybean; BL_4_—brassinolide application at seedling + flowering growth stages of soybean; BL_5_—brassinolide application at seedling + flowering + podding growth stages of soybean; S_0_—1.10 mM/L (control); S_1_—32.40 mM/L; S_2_—60.60 mM/L; S_3_—86.30 mM/L; WAS—weeks after sowing.

**Table 2 plants-10-00541-t002:** Main effects of salinity (S) and brassinolide (BL) on calcium (Ca), potassium (K), magnesium (Mg), sodium (Na) and K^+^/Na^+^ in leaves, stems and roots of soybean at harvest.

Sources of Variance	Calcium (mg/g)			Potassium (mg/g)			Magnesium (mg/g)			Sodium (mg/g)			K^+^/Na^+^		
Salinity (S) Concentrations (mM/L)	Leaves	Stems	Roots	Leaves	Stems	Roots	Leaves	Stems	Roots	Leaves	Stems	Roots	Leaves	Stems	Roots
**S_0_**	3.00 ^a^	0.75 ^a^	0.58 ^a^	22.50 ^a^	14.88 ^a^	15.59 ^a^	1.16 ^a^	0.78 ^a^	0.65 ^a^	0.73 ^a^	0.99 ^c^	4.71 ^b^	66.76 ^a^	21.95 ^a^	5.86 ^a^
**S_1_**	2.36 ^b^	0.54 ^b^	0.39 ^b^	20.39 ^a^	12.59 ^a^	10.20 ^b^	1.00 ^b^	0.70 ^ab^	0.30 ^b^	2.62 ^b^	10.70 ^b^	14.09 ^a^	12.30 ^b^	2.14 ^b^	0.71 ^b^
**S_2_**	2.09 ^c^	0.51 ^bc^	0.32 ^b^	17.59 ^b^	11.22 ^bc^	5.55 ^c^	0.90 ^c^	0.61 ^bc^	0.13 ^c^	5.52 ^a^	15.47 ^a^	14.64 ^a^	7.20 ^b^	0.73 ^b^	0.38 ^b^
**S_3_**	1.39 ^d^	0.47 ^c^	0.30 ^b^	12.39 ^c^	10.06 ^c^	3.81 ^c^	0.81 ^d^	0.53 ^c^	0.09 ^c^	6.29 ^a^	16.28 ^a^	14.90 ^a^	3.37 ^b^	0.62 ^b^	0.25 ^b^
Growth stages of BL applications															
**BL_0_**	1.27 ^d^	0.50 ^b^	0.26 ^a^	11.34 ^b^	7.29 ^c^	3.50 ^c^	0.84 ^b^	0.53 ^b^	0.11 ^c^	5.69 ^a^	11.65 ^a^	13.58 ^a^	8.25 ^b^	2.69 ^a^	0.31 ^b^
**BL_1_**	2.17 ^c^	0.55 ^ab^	0.38 ^a^	17.63 ^a^	11.54 ^b^	8.05 ^b^	0.98 ^a^	0.66 ^ab^	0.31 ^b^	2.88 ^b^	11.13 ^a^	12.15 ^b^	20.56 ^ab^	8.22 ^a^	1.09 ^b^
**BL_2_**	2.28 ^bc^	0.58 ^ab^	0.38 ^a^	19.29 ^a^	12.47 ^ab^	9.26 ^ab^	0.98 ^a^	0.69 ^a^	0.28 ^b^	3.93 ^ab^	10.97 ^a^	11.94 ^b^	22.68 ^ab^	4.83 ^a^	1.27 ^b^
**BL_3_**	2.26 ^bc^	0.57 ^ab^	0.40 ^a^	19.25 ^a^	13.21 ^ab^	9.24 ^ab^	0.98 ^a^	0.69 ^a^	0.29 ^b^	4.19 ^ab^	11.80 ^a^	11.88 ^b^	24.58 ^ab^	7.23 ^a^	1.84 ^b^
**BL_4_**	2.53 ^ab^	0.57 ^ab^	0.43 ^a^	20.75 ^a^	14.06 ^ab^	10.44 ^ab^	1.00 ^a^	0.72 ^a^	0.36 ^ab^	3.18 ^b^	10.54 ^a^	11.64 ^b^	28.55 ^a^	5.83 ^a^	1.78 ^b^
**BL_5_**	2.73 ^a^	0.63 ^a^	0.53 ^a^	21.05 ^a^	14.56 ^a^	12.22 ^a^	1.03 ^a^	0.65 ^ab^	0.42 ^a^	2.86 ^b^	9.07 ^b^	11.30 ^b^	29.81 ^a^	9.37 ^a^	4.55 ^a^

Mean pairs within a column with different letters are significantly different at the 5% probability level according to Duncan’s new multiple-range test. BL_0_—no brassinolide application (control); BL_1_—brassinolide application at seedling growth stage of soybean; BL_2_—brassinolide application at flowering growth stage of soybean; BL_3_—brassinolide application at podding growth stage of soybean; BL_4_—brassinolide application at seedling + flowering growth stages of soybean; BL_5_—brassinolide application at seedling + flowering + podding growth stages of soybean; S_0_—1.10 mM/L (control); S_1_—32.40 mM/L; S_2_—60.60 mM/L; S_3_—86.30 mM/L.

**Table 3 plants-10-00541-t003:** Main and interaction effects of salinity (S) and brassinolide (BL) on leaf dry weight, stem dry weight, harvest index, shoot-to-root ratio and protein content of soybean seeds at harvest.

Sources of Variance	Leaf Dry Weight (g/Plant)	Stem Dry Weight (g/Plant)	Harvest Index (%)	Shoot to Root Ratio	Protein Concentration (%)
**Salinity (S)** **Concentrations (mM/L)**					
S_0_	8.59 ^a^	5.81 ^a^	32.86 ^a^	15.99 ^a^	39.41 ^a^
S_1_	7.58 ^b^	5.15 ^b^	29.78 ^ab^	12.90 ^b^	31.15 ^b^
S_2_	3.26 ^c^	2.37 ^c^	28.48 ^b^	10.79 ^c^	27.53 ^c^
S_3_	2.20 ^d^	1.28 ^d^	19.06 ^c^	9.22 ^d^	20.30 ^d^
**Growth Stages of** **BL Applications**					
BL_0_	2.86 ^e^	2.20 ^c^	18.84 ^b^	8.58 ^d^	15.17 ^c^
BL_1_	4.80 ^d^	3.68 ^b^	27.78 ^a^	11.58 ^c^	30.35 ^b^
BL_2_	5.36 ^cd^	3.95 ^ab^	29.55 ^a^	12.12 ^bc^	29.21 ^b^
BL_3_	6.98 ^a^	4.36 ^a^	30.61 ^a^	12.68 ^b^	32.12 ^b^
BL_4_	6.30 ^ab^	3.73 ^b^	29.95 ^a^	13.04 ^b^	32.68 ^b^
BL_5_	6.13 ^bc^	3.98 ^ab^	28.54 ^a^	15.36 ^a^	38.06 ^a^
**S × BL**					
S_0_ BL_0_	4.73 ^f^	3.85 ^d^	20.06 ^b–e^	10.93 ^f–i^	28.70 ^a^
S_1_ BL_0_	3.40 ^fgh^	2.53 ^ef^	23.35 ^cde^	9.78 ^hij^	19.62 ^a^
S_2_ BL_0_	2.09 ^hi^	1.75 ^f–j^	25.96 ^b–e^	8.14 ^j^	12.37 ^a^
S_3_ BL_0_	1.21 ^i^	0.66 ^j^	0.00 ^f^	5.46 ^k^	0.00 ^a^
S_0_ BL_1_	7.07 ^e^	5.68 ^abc^	30.02 ^a–e^	14.19 ^cd^	37.08 ^a^
S_1_ BL_1_	6.54 ^e^	5.18 ^c^	31.20 ^a–d^	11.96 ^e–h^	33.88 ^a^
S_2_ BL_1_	3.25 ^fgh^	2.81 ^ef^	29.54 ^a–e^	11.02 ^f–h^	28.17 ^a^
S_3_ BL_1_	2.35 ^ghi^	1.03 ^h–j^	20.36 ^e^	9.15 ^ij^	22.27 ^a^
S_0_ BL_2_	7.98 ^cde^	6.44 ^ab^	33.09 ^abc^	15.70 ^bc^	39.40 ^a^
S_1_ BL_2_	7.49 ^de^	5.62 ^abc^	32.83 ^abc^	12.61 ^d–g^	27.60 ^a^
S_2_ BL_2_	3.21 ^fgh^	2.07 ^e–h^	28.34 ^b–e^	10.93 ^f–i^	27.03 ^a^
S_3_ BL_2_	2.77 ^ghi^	1.72 ^f–j^	23.93 ^cde^	9.25 ^ij^	22.84 ^a^
S_0_ BL_3_	11.49 ^a^	6.57 ^a^	39.45 ^a^	17.01 ^b^	40.35 ^a^
S_1_ BL_3_	9.72 ^b^	5.83 ^abc^	30.17 ^a–e^	13.24 ^de^	31.59 ^a^
S_2_ BL_3_	3.97 ^fg^	3.07 ^de^	30.65 ^a–d^	10.48 ^ghi^	32.35 ^a^
S_3_ BL_3_	2.73 ^ghi^	1.98 ^e–i^	22.17 ^de^	9.99 ^hij^	24.17 ^a^
S_0_ BL_4_	10.42 ^ab^	6.19 ^abc^	35.13 ^ab^	16.92 ^b^	40.51 ^a^
S_1_ BL_4_	8.94 ^bcd^	5.41 ^bc^	32.06 ^a–d^	14.06 ^cd^	34.26 ^a^
S_2_ BL_4_	3.66 ^fgh^	2.38 ^efg^	28.52 ^b–e^	11.31 ^e–i^	30.26 ^a^
S_3_ BL_4_	2.16 ^ghi^	0.93 ^ij^	24.10 ^cde^	9.87 ^hij^	25.69 ^a^
S_0_ BL_5_	9.85 ^ab^	6.10 ^abc^	33.40 ^abc^	21.21 ^a^	50.43 ^a^
S_1_ BL_5_	9.37 ^bc^	6.33 ^ab^	29.04 ^b–e^	15.74 ^bc^	39.96^a^
S_2_ BL_5_	3.37 ^fgh^	2.15 ^efg^	27.89 ^b–e^	12.89 ^def^	35.02 ^a^
S_3_ BL_5_	1.94 ^hi^	1.34 ^g–j^	23.82 ^cde^	11.60 ^e–h^	26.84 ^a^

Mean pairs within a column with different letters are significantly different at the 5% probability level according to Duncan’s new multiple-range test. BL_0_—no brassinolide application (control); BL_1_—brassinolide application at seedling growth stage of soybean; BL_2_—brassinolide application at flowering growth stage of soybean; BL_3_—brassinolide application at podding growth stage of soybean; BL_4_—brassinolide application at seedling + flowering growth stages of soybean; BL_5_—brassinolide application at seedling + flowering + podding growth stages of soybean; S_0_—1.10 mM/L (control); S_1_—32.40 mM/L; S_2_—60.60 mM/L; S_3_—86.30 mM/L.

**Table 4 plants-10-00541-t004:** Main and interaction effects of salinity (S) and brassinolide (BL) on number of days to 50% flowering and podding of soybean and number of branches, pods and seeds per plant at harvest.

Sources of Variance	Number of Branches/Plant	Days to 50% Flowering	Days to 50% Podding	Number of Pods/Plant	Number of Seeds/Plant
**Salinity (S)** **Concentrations (mM/L)**					
**S_0_**	25.17 ^a^	19.94 ^d^	35.00 ^c^	35.39 ^a^	88.22 ^a^
**S_1_**	20.06 ^b^	20.78 ^c^	34.44 ^c^	24.39 ^b^	60.44 ^b^
**S_2_**	16.33 ^c^	28.00 ^b^	42.00 ^b^	18.50 ^c^	46.17 ^c^
**S_3_**	10.00 ^d^	34.67^a^	48.67 ^a^	6.00 ^d^	13.78 ^d^
**Growth Stages of BL** **Applications**					
**BL_0_**	13.25 ^d^	27.33 ^a^	43.25 ^a^	15.25 ^b^	35.92 ^b^
**BL_1_**	16.08 ^c^	25.25 ^b^	38.83 ^c^	21.58 ^a^	53.67 ^a^
**BL_2_**	18.42 ^b^	25.33 ^b^	39.00 ^bc^	21.83 ^a^	54.33 ^a^
**BL_3_**	19.25 ^ab^	25.75 ^b^	39.75 ^bc^	22.58 ^a^	56.25 ^a^
**BL_4_**	20.00 ^ab^	25.42 ^b^	39.33 ^bc^	21.92 ^a^	54.67 ^a^
**BL_5_**	20.33 ^a^	26.00 ^b^	40.00 ^b^	23.25 ^a^	58.08 ^a^
**S × BL**					
**S_0_ BL_0_**	19.67 ^a^	23.00 ^c^	44.67 ^b^	25.53 ^a^	63.00 ^a^
**S_1_ BL_0_**	16.00 ^a^	23.67 ^c^	37.67 ^d^	20.67^a^	50.00 ^a^
**S_2_ BL_0_**	13 33 ^a^	27.67 ^b^	41.67 ^c^	12.33 ^a^	30.67 ^a^
**S_3_ BL_0_**	4.00 ^a^	35.00 ^a^	49.00 ^a^	2.67 ^a^	0.00 ^a^
**S_0_ BL_1_**	23.00 ^a^	19.00 ^e^	32.67^fg^	36.33 ^a^	90.67 ^a^
**S_1_ BL_1_**	16.67 ^a^	19.00 ^e^	31.67 ^g^	24.67 ^a^	61.33 ^a^
**S_2_ BL_1_**	15.33 ^a^	28.67 ^b^	42.67 ^c^	20.33 ^a^	50.67 ^a^
**S_3_ BL_1_**	9.33 ^a^	34.33 ^a^	48.33 ^a^	5.00 ^a^	12.00 ^a^
**S_0_ BL_2_**	25.00 ^a^	19.00 ^e^	32.67 ^fg^	36.67 ^a^	91.33 ^a^
**S_1_ BL_2_**	20.67 ^a^	19.33 ^e^	32.33 ^fg^	25.33 ^a^	63.00 ^a^
**S_2_ BL_2_**	16.67 ^a^	28.33 ^b^	42.33 ^c^	19.33 ^a^	48.33 ^a^
**S_3_ BL_2_**	11.33 ^a^	34.67 ^a^	48.67 ^a^	6.00 ^a^	14.67 ^a^
**S_0_ BL_3_**	27.33 ^a^	20.00 ^de^	33.67 ^efg^	38.33 ^a^	95.67 ^a^
**S_1_ BL_3_**	22.33 ^a^	20.67 ^de^	35.00 ^e^	26.33 ^a^	65.67 ^a^
**S_2_ BL_3_**	16.33 ^a^	27.33 ^b^	41.33 ^c^	18.33^a^	45.67 ^a^
**S_3_ BL_3_**	11.00 ^a^	35.00 ^a^	49.00 ^a^	7.33 ^a^	18.00 ^a^
**S_0_ BL_4_**	28.00 ^a^	19.00 ^e^	32.67 ^fg^	34.67 ^a^	86.67 ^a^
**S_1_ BL_4_**	22.33 ^a^	20.33 ^de^	34.33 ^ef^	24.67 ^a^	61.33 ^a^
**S_2_ BL_4_**	17.67 ^a^	28.33 ^b^	42.33 ^c^	20.00 ^a^	50.00 ^a^
**S_3_ BL_4_**	12.00 ^a^	34.00 ^a^	48.00 ^a^	8.33 ^a^	20.67 ^a^
**S_0_ BL_5_**	28.00 ^a^	19.67 ^de^	33.67 ^efg^	41.00 ^a^	102.00 ^a^
**S_1_ BL_5_**	22.33 ^a^	21.67 ^cd^	35.67 ^e^	24.67 ^a^	61.33 ^a^
**S_2_ BL_5_**	18.67 ^a^	27.67 ^b^	41.67 ^c^	20.67 ^a^	51.67 ^a^
**S_3_ BL_5_**	12.33 ^a^	35.00 ^a^	49.00 ^a^	6.67 ^a^	17.33 ^a^

Mean pairs within a column with different letters are significantly different at the 5% probability level according to Duncan’s new multiple-range test. BL_0_—no brassinolide application (control); BL_1_—brassinolide application at seedling growth stage of soybean; BL_2_—brassinolide application at flowering growth stage of soybean; BL_3_—brassinolide application at podding growth stage of soybean; BL_4_—brassinolide application at seedling + flowering growth stages of soybean; BL_5_—brassinolide application at seedling + flowering + podding growth stages of soybean; S_0_—1.10 mM/L (control); S1—32.40 mM/L; S_2_—60.60 mM/L; S_3_—86.30 mM/L.

## Data Availability

Data sharing is not applicable to this article.

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
