# Peer review of "Salinity Effects on Morpho-Physiological and Yield Traits of Soybean (Glycine max L.) as Mediated by Foliar Spray with Brassinolide"

_plants, 2021, doi:10.3390/plants10030541_

Round 1
Reviewer 1 Report
The manuscript addresses an important problem of the effects of salinity on soybean growth and yielding and the possibility of mitigating them through the use of brassinosteroids at various stages of plant development. The work is interesting, but presented in a very inaccessible form to the reader. The authors included too many results. For this reason, I propose changes that could make the text more intelligible.
- I suggest you transfer this detailed data to supplementary files. Leave in the manuscript only those data that really show significant changes. For example, do not give all the observation dates (WAS), there were 8 measurements date, but choose only, for example, three to four maximum. Tables should be written horizontally so that the results do not overlap.
- The authors give statistical differences between all the data, which in my opinion is unnecessary. It is enough to provide these differences only for each combination salt concentration and the hormone used.
- The graphs should be standardized, i.e. the color of the bar in each graph should be the same for a given salt concentration / hormone combination. The stripes and dashes make reading data very tiring.
- The work should end with conclusions that would clearly summarize the most important results.
- Legend of charts and tables. The equal sign between the abbreviations and the explanation should be replaced with a hyphen. There is an error in specifying BL5 everywhere, between flowering and podding you have to replace the equal sign with a plus.
Author Response
Please see attachment: (Response to reviewer 1/cover letter)

Reviewer 2 Report
The authors report a study on the evaluation of the effects of a treatment of Glycine max L. with Brassinolide in conditions of excessive salinity. The authors evaluate different parameters such as leaf morphology, the yield in terms of nutrients, the production and activity of enzymes such as SOD and APX.
The manuscript is an important study on a culture often subjected to significant saline stresses.
However some observations need to be resolved:
1) Since the brassinosteroid class includes many molecules, the authors should clarify which brassinosteroid they used. 24-epibrassinolide?
2) The authors should better explain the choice of dosages for Brassinolide and NaCl. On what rationale is the choice of dosages based?
3) Did the authors observe variations in the content of secondary metabolites in the different experimental groups?
4) I suggest the inclusion of some interesting works:
Reza Yousefi, A. et al. Germination and Seedling Growth Responses of Zygophyllum fabago, Salsola kali L. and Atriplex canescens to PEG-Induced Drought Stress. Environments 2020,
Mahdavi, A. et al., Variation in Terpene Profiles of Thymus vulgaris in Water Deficit Stress Response. Molecules 2020,
Authors should follow the formatting of the text as required by PLANTS. They should also improve the quality of the figures.
Author Response
Please see attachment: (Response to reviewer 2/cover letter)

Reviewer 3 Report
I would like to recommend the present revised version to publish in Plants.